# Influence of Sintering Conditions and Nanosilicon Carbide Concentration on the Mechanical and Thermal Properties of Si_3_N_4_-Based Materials

**DOI:** 10.3390/ma16052079

**Published:** 2023-03-03

**Authors:** Magdalena Gizowska, Milena Piątek, Krzysztof Perkowski, Agnieszka Antosik

**Affiliations:** Łukasiewicz Research Network, Institute of Ceramics and Building Materials, 8 Cementowa Street, 31-983 Krakow, Poland

**Keywords:** fracture toughness, hardness, hot isostatic pressing, silicon carbide nanopowder, silicon nitride, thermal conductivity

## Abstract

In the work, silicon nitride ceramics (Si_3_N_4_) and silicon nitride reinforced by nano silicon carbide particles (Si_3_N_4_-nSiC) in amounts of 1–10 wt.% were investigated. The materials were obtained using two sintering regimes: under conditions of ambient and high isostatic pressure. The influence of the sintering conditions and the concentration of nanosilicon carbide particles on the thermal and mechanical properties was studied. The presence of highly conductive silicon carbide particles caused an increase in thermal conductivity only in the case of the composites containing 1 wt.% of the carbide phase (15.6 W·m^−1^·K^−1^) in comparison with silicon nitride ceramics (11.4 W·m^−1^·K^−1^) obtained under the same conditions. With the increase in the carbide phase, a decrease in the densification efficiency during sintering was observed, which caused a decrease in thermal and mechanical performance. The sintering performed using a hot isostatic press (HIP) proved to be beneficial in terms of mechanical properties. The one-step high-pressure assisted sintering process in the HIP minimizes the formation of defects at the sample surface.

## 1. Introduction

Due to the excellent properties of silicon nitride, it finds applications as high-performance ceramics. The two main branches of silicon nitride applications are wear-resistant materials (e.g., bioceramics, cutting tools, bearings) and high-temperature materials (e.g., crucibles, thermocouple tubes). The combination of high strength, low thermal expansion, and thermal and chemical resistance means that silicon nitride material can be used in combustion chambers for rocket nozzles [1]. The high working temperatures, thermal shock resistance, and resistivity to oxidation can be tailored by careful control of the introduced sintering additives, the porosity, and the presence of a second phase [2,3,4,5].

Silicon nitride particulate composites have been investigated in terms of improvements in fracture toughness and creep behavior [4,5,6,7]. Most works focused on hot-pressed materials. The hot pressing technique guarantees high densification of multiphase materials; however, it can only be applied for the fabrication of elements with limited dimensions and shapes. In the case of the production of bulk elements with complicated shapes, pressureless sintering is usually considered, which leads to increased porosity with increased concentration of the second phase in silicon-nitride-based composites [8]. There are few reports concerning the sintering of silicon nitride materials in a hot isostatic press (HIP). The research proves that the sintering of silicon nitride in a hot isostatic press leads to a decrease in fracture toughness and an increase in mechanical strength [9]. 

Composite materials in the Si_3_N_4_-SiC system were previously obtained using the HIP technique in a post-sintering process [10]. No references were found on the efficiency of consolidation of the Si_3_N_4_-SiC materials in a HIP in a one-step sintering process.

In the presented paper, the influence of the sintering conditions and the concentration of SiC particles in the Si_3_N_4_ matrix on physical, thermal, and mechanical properties was investigated.

## 2. Materials and Methods

In this research, silicon nitride granulate Starceram N rtp, grade M provided by HC Starck Ceramics GmbH (Selb, Germany), and silicon carbide nanopowder provided by ABCR GmbH (Karlsruhe, Germany) were used.

The samples were prepared from silicon nitride granulate and granulate with 1, 5, and 10 wt.% addition of silicon carbide nanopowder (nSiC). The silicon nitride granulate contains sintering additives (mainly yttria and alumina) and technological binders and plasticizers, which increase the densification effectivity during forming. The granulates containing nanosilicon carbide were prepared by mixing silicon nitride granulate with an appropriate amount of silicon carbide nanopowder in propanol in a ball mill via planetary milling for 1 h. Silicon nitride balls of 3 mm in diameter were used for the mixing of the powders.

The granulates were uniaxially pressed in molds with zirconia liner into cylindrical samples with a diameter of about 16 mm and height of 2.2 mm for the flexural strength test and 3.8 mm for other tests. The samples were subsequently isostatically densified in a cold isostatic press at a pressure of 200 MPa. The green bodies, placed in a graphite crucible, were then sintered in two regimes: (1) pressureless sintering in nitrogen flow in a graphite furnace at a temperature of 1800 °C; (2) hot isostatic pressing (HIP) with a sintering temperature of 1800 °C and pressure of 200 MPa of nitrogen. In both furnaces, nitrogen of 5.0 purity was used.

The density and open porosity of the sintered bodies were measured by the Archimedes method. The measurements were conducted in water. The results were averaged, and the standard deviation was calculated based on the measurements of at least ten samples. The relative density is the ratio between the apparent and theoretical density. The theoretical densities of each composition were calculated according to the rule of mixtures.

The particle size was characterized by the dynamic light scattering technique (dls). Particle size distribution analysis was conducted by means of the zeta potential and a Zetasizer Nano ZS particle size analyzer (Malvern Instruments Ltd., Worcestershire, UK) for diluted suspensions, which underwent ultrasonication (VibraCell VCX130, Sonics & Materials, Inc., Newtown, CT, USA) prior to measurement. The analysis results are presented in terms of the Z average (Z_ave_, also referred to as the cumulant mean or harmonic intensity averaged particle diameter) and polydispersity index (PdI); these data were derived from the intensity of the overall mean particle diameter value and the overall average polydispersity, respectively.

The specific surface area was measured according to the Brunauer–Emmett–Teller (BET) theory using a gas adsorption analyzer (Gemini VII, Micromeritics Instrument Corp., Norcross, GA, USA). Based on the results of the BET, the equivalent spherical particle diameter (d_BET_) was calculated.

Imaging and energy-dispersive spectroscopy (EDS) analyses were performed by means of scanning electron microscopy (SEM; Nova NanoSEM 200, FEI Company, Hillsboro, OR, USA). For the EDS, three areas were analyzed, from which 3–5 points were selected for measurement.

The X-ray diffraction (XRD) was performed in a Bragg–Brentano system using a Bruker-AXS D8 DAVINCI diffractometer equipped with a copper X-ray source. The XRD analysis was performed on the unpolished surfaces. Diffractograms were recorded in the angular range of 5 to 120° 2θ (Cu Kα, with radiation wavelength 0.154 nm), the measurement step was 0.01°, and the time was 2 sec per step. The optical system of the diffractometer consisted of a 0.3° divergence slit, a 1.5° anti-scattering slit, two 2.5° Soller slits, a Ni filter, and a LynxEye strip detector with a field of view of 2.94°. Identification of the phases was performed by comparing the recorded diffractograms with patterns found in the COD database using the DIFFRACplus EVA-SEARCH program. Quantitative X-ray analysis was performed based on published crystal structures (COD) using the Rietveld method in Topas v5.0 software [11,12].

The hardness of the materials obtained was determined by the Vickers method using a LECO LV800 hardness tester. The measurement was performed according to the standards PN-EN ISO 14705:2021-06 and PN-EN ISO 6507-1:2018-05. Vickers pyramid impressions were made using a force of 1 kgf on the surface of the polished samples. The rate of movement of the pyramid was 0.2 mm/min. After reaching the set force, the load was maintained for 15 s.

The indentation fracture resistance was calculated based on Palmquist cracks (0.25 < l/a < 1.5) propagating from the apex of the Vickers indent using Equation (1) [13,14,15]:(1)$$K_{IFR}=0.018\cdot {\left(\frac{E}{H_{v}}\right)}^{0.4}\cdot {\left(\frac{l}{a}\right)}^{-0.5}\cdot H_{v}\cdot \sqrt{a}$$
where *a* is half of the diagonal of the indentation induced by the Vickers indenter (a = 0.5 d), *H_V_* is the Vickers hardness, *E* is Young’s modulus, and *c* is the length of the Palmquist crack. The length of the crack was measured in a straight line from the apex of the indent to the tip of the crack.

In order to determine the flexural strength, the ball-on-ring testing method was used, which involves measuring the maximum failure force of cylindrical specimens laid flat on a ring-shaped stand and loaded axially. The tests were performed using a Tinius Olsen H10-KS testing machine. The flexural strength index using the biaxial loading method was determined from the Kirstein and Wooley equation [16,17]:(2)$$\sigma_{max}=\frac{3P\left(1-v\right)}{4\pi t^{2}}\cdot \left[1+2ln\frac{a}{b}+\frac{\left(1-v\right)}{\left(1+v\right)}\left\{1-\frac{b^{2}}{2a^{2}}\right\}\frac{a^{2}}{R^{2}}\right]$$
where P is the failure load [N], t is the thickness of the disk, a is the radius of the supporting ring (12 mm), b is the radius of uniform load (b ≈ t/3), R is the radius of the disk, and ν is Poisson’s ratio for Si_3_N_4_ (v[Si_3_N_4_] = 0.255 was used for calculations).

The choice of the ball-on-ring technique over the 3- and 4-point flexural strength was motivated by several reasons. First of all, in this test, the friction between the tested sample and the jig is minimized. Secondly, the stresses usually present at the edges of rectangular samples have a great influence on the 3-point or 4-point flexural strength test results. In the case of the ball-on-ring technique, this phenomenon is avoided [18]. Last but not least, test samples in the cylindrical shape are easy to prepare, and their geometry introduces fewer defects related to shrinkage stresses into the ceramic body. The flexural tests were conducted on “as received” samples.

The value of the Weibull modulus was estimated for both the Si_3_N_4_ and Si_3_N_4_-nSiC specimens. The Weibull modulus was determined from the results of a flexural strength test by a graphical method using Equation (3):(3)$$\ln\left(ln\frac{1}{1-F}\right)=m\;ln\sigma +C$$
where *F* is the probability of brittle decohesion of the specimen, m is the Weibull modulus, σ is the flexural strength, and *C* is the constant resulting from the double logarithmic operation of the basic expression relating the probability of failure of a shape to its strength:(4)$$F=1-e^{-{\left(\frac{\sigma -\sigma_{u}}{\sigma_{0}}\right)}^{m}}$$
where σ_u_ is the parameter of asymmetry of the distribution of strength values σ, and σ_0_ is a normalizing constant [19,20].

The thermal diffusivity measurement was carried out by the LaserFlash (LFA) technique using an LFA 427 analyzer (Netzsch GmbH). In this method, the lower surface of the sample placed in a holder is submitted to a short energy pulse produced by a neodymium laser. This results in a temperature change of the second, upper surface of the plane parallel surface. The temperature changes measured by an infrared detector, type InSb, are then registered as a function of time. The thermal diffusivity was calculated from Equation (5) given by Parker et al. [21]:(5)$$\alpha =0.1388\cdot \frac{L^{2}}{t_{1/2}}$$
where α is the thermal diffusivity, L is the sample thickness, and t_1/2_ is the half-time (time value at half the temperature signal height).

The thermal conductivity was calculated based on Equation (6) [22]:(6)$$\lambda =d\cdot c_{p}\cdot \alpha$$
where λ is the thermal conductivity, d is the density of the measured sample, and c_p_ is the specific heat of the material.

## 3. Results

The starting materials were characterized in terms of their particle size (Table 1). Additionally, the silicon carbide nanopowder was studied by the SEM technique (Figure 1).

According to the specific surface measurements results, the Si_3_N_4_ powder, which was the basis of the used granulate, was of submicrometric size (d_BET_ = 135 nm); however, in the dls measurements, the presence of agglomerates was detected (Z_ave_ = 1.1 μm).

The silicon carbide nanopowder also showed some agglomeration in the dls measurement, as the cumulant mean was higher than the BET equivalent particle diameter (Z_ave_ = 397 nm, d_BET_ = 66 nm). In the SEM image, it was observed that most of the grains were of nanometric size (<100 nm), and there were few grains of submicrometric size visible (100–400 nm; Figure 1).

Such a combination of granulometric sizes of the powders should provide good densification in a green state, as the green density of the silicon nitride–silicon carbide mixture of powders was higher than the density of silicon nitride green samples. The green relative density for silicon nitride samples was 60.5%, and the value measured for the Si_3_N_4_-nSiC samples was in the range of 61.3–62.3% (Table 2).

The powders were investigated in terms of their phase composition (Figure 2 and Figure 3).

The silicon nitride granulate consisted mainly of the α-Si_3_N_4_ phase (78.7 wt.%). The β-Si_3_N_4_ phase was also detected, and its concentration was 12.4 wt.%. The oxide phases Y_2_O_3_ and α-Al_2_O_3_ were the sintering agents, and they were present in an amount of about 4.4 wt.% each.

The silicon carbide nanopowder consisted mainly of the β-SiC phase (ca. 70%), which is a cubic crystallographic form of silicon carbide. The rest of the material was the hexagonal α-SiC.

The samples sintered at a temperature of 1800 °C were investigated in terms of their phase composition. The results of the XRD analysis of the Si_3_N_4_ sample and the sample containing 5 wt.% of nanosilicon carbide, together with the characteristic reflections of the phases present in the samples, are presented in Figure 4 and Figure 5.

In the samples sintered at a temperature of 1800 °C, silicon nitride occurred predominantly in the form of β-Si_3_N_4_. XRD analysis performed on a polished sample led to the detection of only β-Si_3_N_4_. The α-Si_3_N_4_ present in the granulate is more reactive and reacts with the oxides, forming SiAlONs, which then form an amorphous phase [7,23]. At the unpolished surface of the sample, SiAlON occurs in a hexagonal phase. Some residual amounts of oxides were detected in the surface layer; their quantity was estimated to be beneath 1%.

The silicon carbide transformed into the hexagonal α-SiC crystalline form. The β-SiC phase is metastable and transforms into α-SiC [24].

There were differences in the phase composition both on the surface and in the sample body of the Si_3_N_4_ and Si_3_N_4_-nSiC materials. At the surface of Si_3_N_4_ ceramics, the amounts of the α and β phase of silicon nitride and SiAlON were 7.7%, 88.8%, and 3.1%, respectively. In the Si_3_N_4_-nSiC material, the mass composition of the mentioned phases was 7.0%, 80.2%, and 12.2%, respectively. Additionally, in the Si_3_N_4_-nSiC material, the measured phase content of SiC was smaller than the amount of introduced material and equaled about 1%. The XRD of the Si_3_N_4_-nSiC material performed on the polished sample revealed that the body consisted mainly of β-Si_3_N_4_ with minor amounts of α-Si_3_N_4_ and hexagonal SiAlON phase. From the analysis, it can be concluded that the silicon carbide reacts with the oxides and is a competitive phase for the reaction with the oxides in relation to α-Si_3_N_4_.

Sintered Si_3_N_4_ ceramics and Si_3_N_4_-nSiC material samples were tested in terms of their density, open porosity, and thermal diffusivity. The results are presented in Table 2.

The theoretical density of silicon nitride and silicon nitride containing 1 wt.% of silicon carbide nanopowder sintered pressureless at a temperature of 1800 °C exceeds 98%. The Si_3_N_4_-nSiC materials containing 5 and 10 wt.% of nSiC sintered in the same regime showed lower densification, and their theoretical density was about 95%. All samples show similar open porosity in the range of 1–1.6%.

Samples sintered via HIP have lower density. The Si_3_N_4_ ceramics and 1 wt.% Si_3_N_4_-nSiC materials were densified in the HIP process to theoretical densities of 93.9 and 94.7%, respectively. The materials containing higher concentrations of nSiC underwent less effective densification, and their theoretical densities were 86.8 and 80.7%, respectively, for bodies containing 5 and 10 wt.% of the second phase. The open porosity of Si_3_N_4_ ceramics and Si_3_N_4_-1 wt.% nSiC bodies was 0.3%. In samples containing 5 and 10 wt.% of nSiC, the open porosity values were 3.7 and 17.5%, respectively.

The thermal conductivity of the bodies sintered in atmospheric pressure was higher than the values calculated for the Si_3_N_4_ and Si_3_N_4_-nSiC materials sintered in the HIP process. The highest thermal conductivity was calculated for the sample sintered in a pressureless regime with 1 wt.% of nSiC. All the HIP samples had similar thermal conductivity in the range of 10–13 W·m^−1^·K^−1^.

The mechanical properties are presented in Table 3.

The flexural strength of the samples sintered under atmospheric pressure was relatively low and was in the range of 220–280 MPa, with the highest value of 276 MPa registered for the silicon nitride sample. The relation between the concentration of nanosilicon carbide and flexural strength is not direct. The value dropped with an increase in nSiC concentration up to 5 wt.% from 238 to 221 MPa, whereas the flexural strength of Si_3_N_4_-nSiC materials containing 10 wt.% of silicon carbide rose to 260 MPa.

The silicon nitride and silicon nitride materials containing 1 and 5 wt.% of nSiC after the HIP process showed much higher flexural strength than those sintered under atmospheric pressure. The flexural strength of Si_3_N_4_ and Si_3_N_4_-1 wt% nSiC samples after pressure-assisted sintering was about 490 MPa. The material containing 5 wt.% of nSiC showed a strength of 416 MPa. With a further increase in the nSiC concentration to 10 wt.%, the flexural strength dropped to 250 MPa.

The Weibull modulus calculated based on the flexural strength test results was in the range of 5.3 to 9.1. The samples sintered under nitrogen at atmospheric pressure did not show a direct correlation between the Weibull modulus and second-phase concentration. The values for silicon nitride and silicon nitride containing 1 wt.% nSiC were similar: 5.4 and 5.3. With an increase in the nSiC content to 5 wt.%, the Weibull modulus rose to 8.3 and dropped again to a value of 6.8 for materials containing 10 wt.% of nSiC. In the case of the samples sintered under conditions of nitrogen at high isostatic pressure, the relation between the nSiC content and Weibull modulus was different. For the silicon nitride, the Weibull modulus was 7.1, but for the material containing 1 wt.% of nSiC, it was much lower and equaled 4.4. With an increase in the nSiC content, an increase in the modulus value was observed: 6.4 and 9.1 for 5 and 10 wt.% Si_3_N_4_-nSiC materials, respectively.

The hardness of the Si_3_N_4_ and Si_3_N_4_-nSiC materials containing 1 and 5 wt.% was about 19 GPa for all the samples sintered under atmospheric pressure. The material with 10 wt.% of nSiC had a lower hardness of 16.1 GPa.

Samples sintered under high isostatic pressure had generally lower hardness, with 16.2 GPa for silicon nitride and 17.1 GPa for Si_3_N_4_-1 wt.% nSiC material. With increasing nSiC content to 5 and 10 wt.%, the hardness dropped to 11.8 and 6.7 GPa.

The indentation fracture resistance of HIP samples was higher than that of samples sintered without external pressure, in the ranges of 4.2–4.4 MPa·m^1/2^ and 3.5–3.7 MPa·m^1/2^, respectively. Due to the high porosity of samples containing 10 wt.% silicon carbide sintered in the HIP process, the measurement of the length of the cracks was impossible.

In Figure 6, micrographs of the silicon nitride ceramics and silicon nitride containing 5 wt.% silicon carbide sintered in nitrogen at atmospheric pressure and in HIP conditions are presented.

According to the EDS analysis (not shown), the dark grey areas in the images of Si_3_N_4_ ceramics (Figure 6a–f) are the silicon nitride grains (indicated with solid arrows). The lighter areas are the bonding phase consisting of Y-Si-Al-O-N and Si-Al-O-N systems (indicated with a dashed arrow). These are the Y-α-SiAlON and Y_2_Si_3_O_3_N_4_ crystalline phases detected in the XRD analysis and amorphous phases, which can also form under these conditions [7]. In the fracture surfaces, longitudinal β-Si_3_N_4_ particles are visible in both the materials sintered under pressureless conditions (Figure 6a–c) and those subjected to the HIP process (Figure 6d–f). The samples obtained by pressure-assisted sintering in the HIP are characterized by smaller grains and higher porosity, which was confirmed by the density measurements (Table 2).

In the cross sections of the Si_3_N_4_-nSiC materials, the contrast between the Si_3_N_4_ matrix and the SiC phase is low due to the small difference in atomic mass of carbon and nitrogen. The SiC phase in the form of submicrometric grains is visible in the SEM image as light-grey areas. There was no delamination between nSiC and the matrix observed. According to the SEM observations and the EDS analysis, the SiC phase was uniformly distributed in the matrix. Both Si_3_N_4_ and SiC grains were surrounded uniformly by the bonding phase constituted of the Y-Si-Al-O-N and Si-Al-O-N systems. Similarly to Si_3_N_4_ ceramics, the pressureless sintered Si_3_N_4_-nSiC materials were less porous, and the grains of the matrix underwent higher grain growth.

There were significant differences in the microstructure of the surfaces of the samples depending on the sintering regime. The pressureless sintered Si_3_N_4_ ceramics (Figure 6c) and Si_3_N_4_-nSiC materials (Figure 6i) had defects at the surface. At the surface, there was some porosity visible, reaching a depth of 60–80 μm (indicated in Figure 6c,i). A greater amount of oxide phase (light areas) was observed in this region as well.

The samples obtained by the HIP process had no defects visible on the sample surface. The fracture surface images taken of the sample surface (Figure 6f,l) did not vary significantly from the images of the middle of the sample (Figure 6e,k).

## 4. Discussion

The physical, mechanical, and thermal properties of the Si_3_N_4_ ceramics and Si_3_N_4_-nSiC materials sintered under nitrogen at atmospheric pressure and under nitrogen at high pressure in the hot isostatic pressing process are presented in the form of radial graphs in Figure 7. Note that the open porosity of the samples is presented in the inverted scale.

The materials sintered in the pressureless regime all had some level of open porosity. Although the density of the material slightly decreased with increasing silicon carbide content, the value of open porosity was in the range of 1.0–1.6%. The relative density of the samples indicates that there should be no open porosity at all, as, depending on the material, the transformation of open porosity to closed porosity occurs after reaching 90–95% theoretical density [25,26,27]. Thus, the measured open porosity must result from defects present at the surface of the sample (Figure 6). These defects could not be avoided by sintering in the Si_3_N_4_ powder bed. The furnace chamber was vacuumed and flushed with nitrogen before the sintering process; nonetheless, at the surface of the powder, some water and air still could have been adsorbed, which reacted with the silicon nitride matrix of the samples. The Si_3_N_4_ powder, in which the samples were buried for the sintering process, itself might have been the source of oxidation of the sample surfaces.

These defects are responsible for low flexural strength. The porosity at the surface might be the origin of the failure of the samples. In the case of materials sintered via HIP, which had no major defects at the surface, the flexural strength was higher, even by over 100%, in the case of Si_3_N_4_-1 wt% nSiC materials in comparison to the pressureless sintered samples. The relation between the surface roughness and the mechanical behavior of silicon nitride was also confirmed by other researchers [28]. Additionally, the increased amount of oxide phases at the surface may also have caused the decrease in flexural strength. The observed fracture surfaces (Figure 6) are typical for this intergranular character. This means that the intergranular SiAlON phase is weaker than the Si_3_N_4_ grains.

The beneficial effect of the presence of silicon carbide on mechanical strength was not confirmed. The materials containing 1 wt.% of nanosilicon carbide were characterized by comparable flexural strength in comparison to the silicon nitride matrix, with values of 487 and 485 MPa, respectively. With increasing silicon carbide content, the flexural strength decreased to 416 and 250 MPa for Si_3_N_4_-5 wt% nSiC and Si_3_N_4_-10 wt% nSiC, respectively. The porosity of the Si_3_N_4_-nSiC materials is one of the factors responsible for this phenomenon. The second is the increase in the amorphous SiAlON phase in the Si_3_N_4_-nSiC materials, which may determine the mechanical strength.

A concentration of nanosilicon carbide above 5 wt.% hindered the densification of the Si_3_N_4_-nSiC material bodies. In the case of the pressureless sintering regime, the density of the materials dropped to about 95%. However, in the case of samples sintered via HIP, the relation between the concentration of the second phase and the density and porosity was more pronounced. The HIP materials containing 10 wt.% of nSiC were poorly densified (d_rel_ = 80.7%) and were porous (P_o_ = 17.5%). The mechanism responsible for the poor densification of the materials is the pinning effect of the SiC phase located at the graifn boundaries hindering densification by grain growth [29,30], as the grains observed in the materials were smaller.

The presence of silicon carbide in the silicon nitride matrix had a positive influence on the increase in thermal conductivity. The improvement was best visible in the material containing 1% silicon carbide nanopowder sintered pressureless, which showed thermal conductivity about 37% higher in comparison to the silicon nitride matrix. With increasing silicon carbide content in the silicon nitride bodies, the thermal conductivity gradually decreased, and the 10 wt.% nSiC materials showed thermal conductivity similar to that of the Si_3_N_4_ body. This effect is attributed to the increasing porosity of the bodies with higher nSiC content. The presence of pores compensates for the effect of the presence of highly conductive nSiC in the matrix.

The presence of silicon carbide in the silicon nitride matrix also influenced the hardness of the material. Despite the decreasing relative density, the hardness was about 19 GPa for the pressureless sintered samples. Only for the material containing 10 wt.% of nSiC was a decrease in hardness observed. In the case of the samples obtained by pressure-assisted sintering, the measured hardness decreased with increasing nSiC content, which is related to the worse densification of the materials.

The presence of the second phase in the silicon nitride matrix did not have much influence on the indentation fracture resistance, equaling 3.5–3.7 MPa·m^1/2^ for the pressureless sintered materials and 4.2–4.4 MPa·m^1/2^ for the HIP materials. It was observed that the cracks propagated at the grain boundaries and were usually terminated at the elongated β-Si_3_N_4_ grains positioned perpendicularly to the cracks. Interaction of the cracks with nSiC grains was not observed. The presence of silicon carbide may even cause a decrease in the indentation fracture resistance of the silicon nitride matrix [7,31]. A beneficial influence of the second phase was detected only in the case of the materials containing coarse silicon carbide particles and whiskers [29,31]. However, the toughening effect was observed only up to a definite amount of the additive, which is attributed to the increased porosity of the materials. A similar effect was observed for the materials sintered in HIP: silicon nitride and silicon nitride materials containing 1 and 5 wt.% of nSiC had K_IFR_ = 4.2–4.4 MPa·m^1/2^, while materials containing 10 wt.% showed porosity, which disabled the measurement of the crack lengths.

The presence of the silicon carbide had a minor effect on the indentation fracture resistance. On the other hand, a significant increase in indentation fracture resistance was observed for materials sintered under HIP conditions. This suggests that this parameter is controlled by the quality of the silicon nitride matrix. According to the literature, the indentation fracture resistance in silicon nitride materials is controlled by the formation of β-Si_3_N_4_ and the intergranular phases [29,32,33]. In both Si_3_N_4_ ceramics and Si_3_N_4_-nSiC, the main phase of the matrix was β-Si_3_N_4_, and the oxide binding phases were the same. The presence of the silicon carbide grains did not affect the formation of the matrix material.

## 5. Conclusions

Pressure-assisted sintering of silicon nitride hindered the formation of defects at the surface of the material and improved the mechanical performance of the as-prepared materials.

The addition of highly conductive silicon carbide particles increased thermal conductivity. However, only small amounts of the second phase brought a desirable effect, as with increasing nSiC content, the effectiveness of densification during the sintering process decreased.

Enhanced thermal conductivity indicated that the silicon carbide particles were well bonded with the silicon nitride matrix, with no cavitations at the phase boundaries, which was also confirmed by the microstructural observations.

## Figures and Tables

**Figure 1 materials-16-02079-f001:**
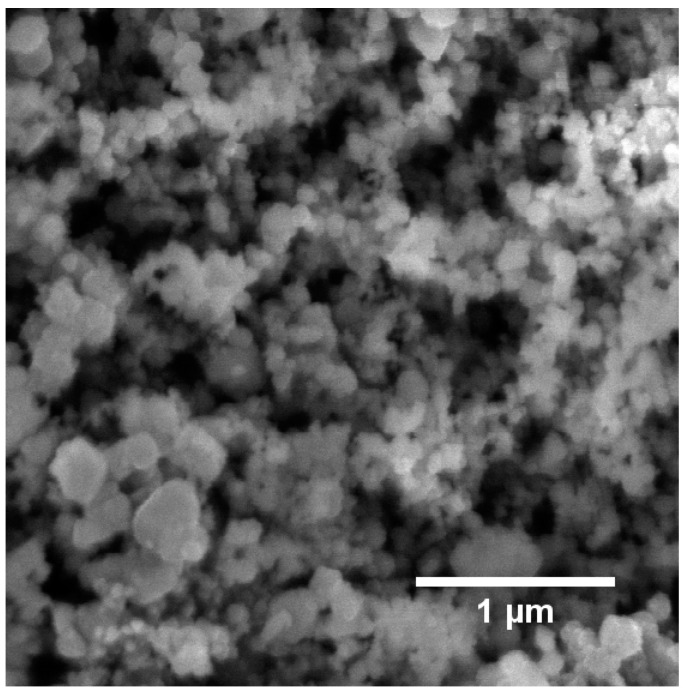
Scanning electron microscopy image obtained via secondary electron detection of the silicon carbide nanopowder.

**Figure 2 materials-16-02079-f002:**
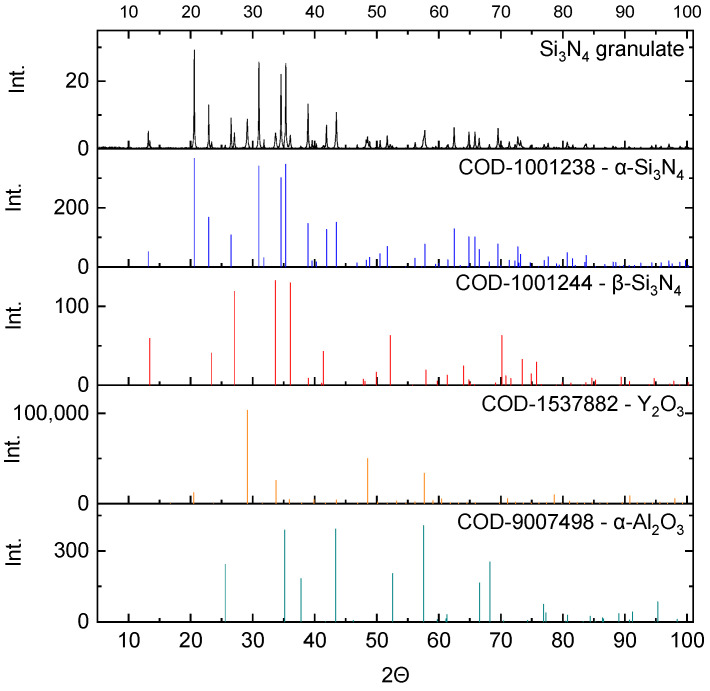
Diffractogram of the Si_3_N_4_ granulate used in the research and the intensity of the characteristic reflections of phases present in the sample.

**Figure 3 materials-16-02079-f003:**
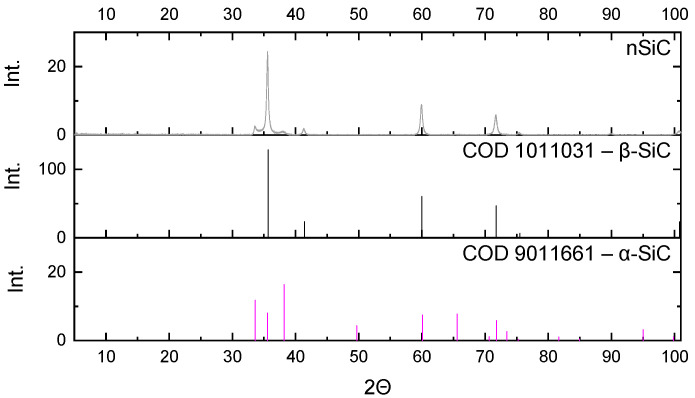
Diffractogram of the silicon carbide nanopowder (nSiC) used in the research and the intensity of the characteristic reflections of phases present in the sample.

**Figure 4 materials-16-02079-f004:**
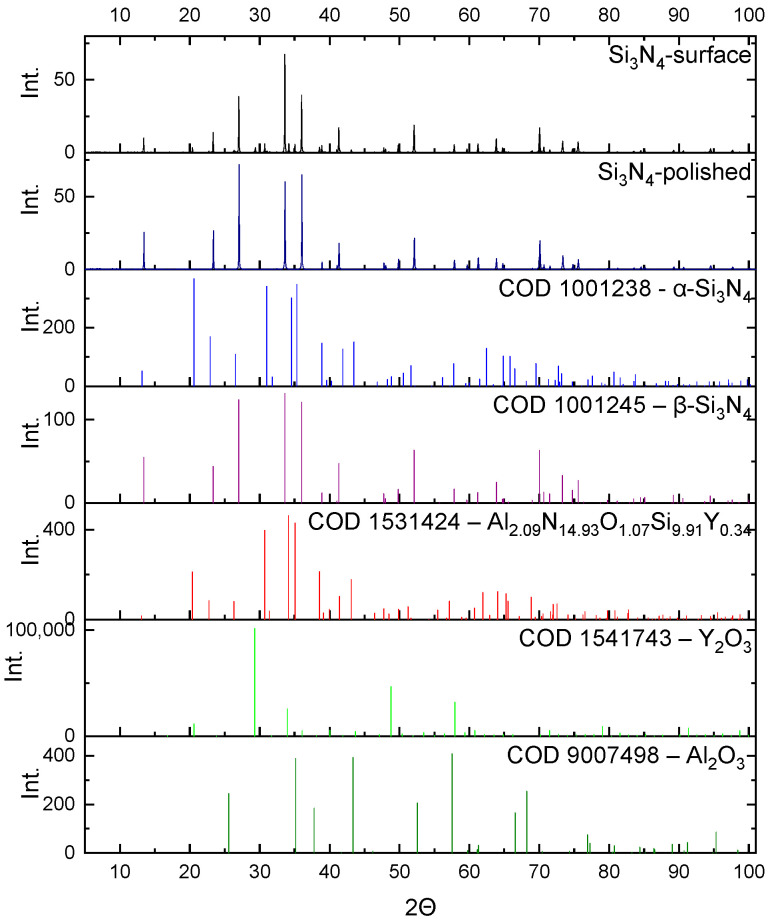
Diffractogram of the Si_3_N_4_ material sintered at a temperature of 1800 °C (HIP) and the intensity of the characteristic reflections of the phases present in the sample.

**Figure 5 materials-16-02079-f005:**
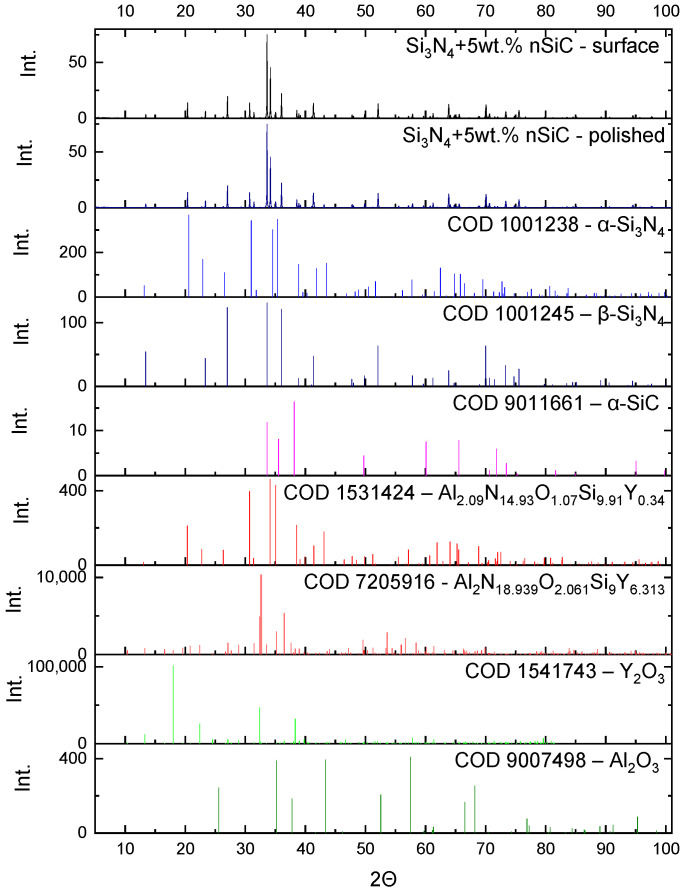
Diffractogram of the Si_3_N_4_-nSiC material containing 5 wt.% of silicon carbide sintered at a temperature of 1800 °C (HIP) and the intensity of the characteristic reflections of the phases present in the sample.

**Figure 6 materials-16-02079-f006:**
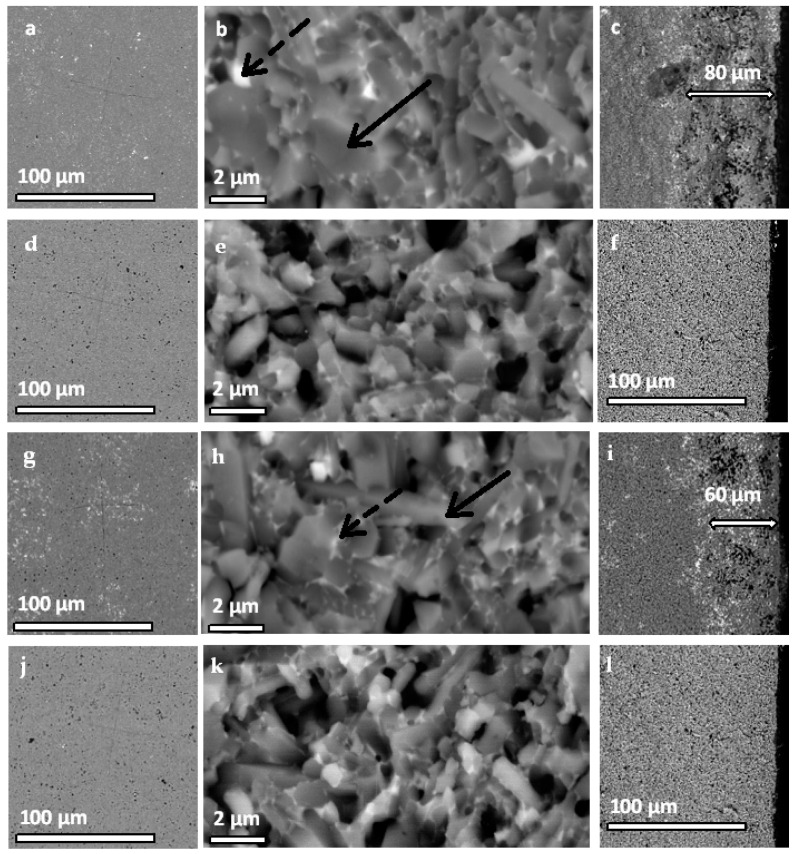
Micrographs of the Si_3_N_4_ material sintered pressureless at a temperature of 1800 °C (**a**–**c**) and in the hot isostatic press at a temperature of 1800 °C (**d**–**f**); Si_3_N_4_-nSiC material containing 5 wt.% of silicon carbide sintered pressureless at a temperature of 1800 °C (**g**–**i**) and in the hot isostatic press at a temperature of 1800 °C (**j**–**l**). In the first column (**a**,**d**,**g**,**j**), the polished surfaces are presented; in the second (**b**,**e**,**h**,**k**), the fracture surfaces of the samples are presented; and in the third column, the fracture surfaces at the edge of the samples are presented (**c**,**f**,**i**,**l**).

**Figure 7 materials-16-02079-f007:**
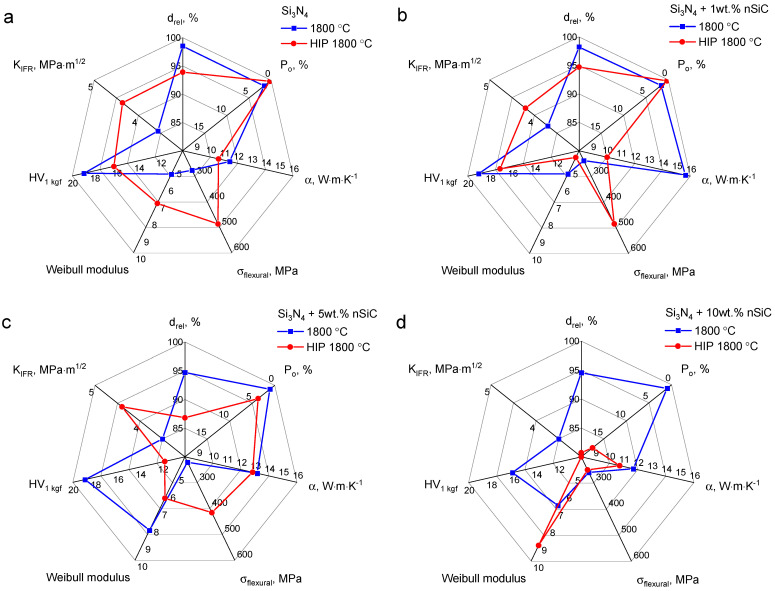
Radial graphs of the selected physical, thermal, and mechanical properties of the Si_3_N_4_ ceramics (**a**), Si_3_N_4_-1 wt.% nSiC (**b**), Si_3_N_4_-5 wt.% nSiC (**c**) and Si_3_N_4_-10 wt.% nSiC (**d**) materials obtained by pressureless sintering (1800 °C) and under conditions of high temperature and high isostatic pressure (HIP 1800 °C).

**Table 1 materials-16-02079-t001:** Properties of the powders used in the research (Z_ave_—cumulant mean, PdI—polydispersity index, S_BET_—specific surface area, d_BET_—BET equivalent spherical particle diameter).

	Z_ave_, nm	PdI, -	S_BET_, m^2^/g	d_BET_, nm
Si_3_N_4_	1063	0.51	14 *	135
nSiC	397	0.26	28.4	66

* data provided by the producer.

**Table 2 materials-16-02079-t002:** Selected physical and thermal properties of the materials were obtained by sintering under ambient pressure at a temperature of 1800 °C and in the hot isostatic press at a temperature of 1800 °C.

Material	Sintering Conditions	Relative Green Density		Relative Density		Open Porosity		Thermal Diffusivity	Thermal Conductivity
		%		%		%		mm^2^·s^−1^	W·m^−1^·K^−1^
Si_3_N_4_	1800 °C	60.5	±0.4	98.5	±0.8	1.6	±3.7	14.0	11.4
	HIP 1800 °C			93.9	±1.4	0.3	±0.1	10.8	10.6
Si_3_N_4_ + 1 wt.% nSiC	1800 °C	62.3	±1.7	98.2	±0.8	1.6	±0.3	13.9	15.6
	HIP 1800 °C			94.7	±0.8	0.3	±0.1	10.3	10.0
Si_3_N_4_ + 5 wt.% nSiC	1800 °C	61.3	±2.0	94.7	±0.6	1.1	±0.2	13.4	13.2
	HIP 1800 °C			86.8	±0.2	3.7	±3.2	11.2	12.8
Si_3_N_4_ + 10 wt.% nSiC	1800 °C	62.1	±2.3	94.6	±0.8	1.0	±0.5	13.2	11.7
	HIP 1800 °C			80.7	±0.3	17.5	±0.2	10.5	10.7

**Table 3 materials-16-02079-t003:** Mechanical properties of the materials obtained by pressureless sintering at a temperature of 1800 °C and in the hot isostatic press at a temperature of 1800 °C.

Material	Sintering Conditions	Flexural Strength		Weibull Modulus	Vickers Hardness		Indentation Fraction Resistance, K_IFR_	
		MPa		-	GPa		MPa·m^1/2^	
Si_3_N_4_	1800 °C	276	±63	5.4	18.9	±2.1	3.6	±0.4
	HIP 1800 °C	487	±67	7.1	16.2	±1.3	4.4	±0.4
Si_3_N_4_ + 1 wt.% nSiC	1800 °C	238	±83	5.3	19.0	±1.5	3.7	±0.3
	HIP 1800 °C	485	±117	4.4	17.1	±0.9	4.2	±0.4
Si_3_N_4_ + 5 wt.% nSiC	1800 °C	221	±57	8.3	18.9	±0.9	3.5	±0.2
	HIP 1800 °C	416	±78	6.4	11.8	±1.4	4.4	±0.6
Si_3_N_4_ + 10 wt.% nSiC	1800 °C	260	±46	6.8	16.1	±1.1	3.5	±0.3
	HIP 1800 °C	250	±33	9.1	6.7	±0.3	-*	

* not applicable.

## Data Availability

Not applicable.

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
