# Peer review of "Influence of Sintering Conditions and Nanosilicon Carbide Concentration on the Mechanical and Thermal Properties of Si3N4-Based Materials"

_materials, 2023, doi:10.3390/ma16052079_

Round 1
Reviewer 1 Report
Gizowska et al investigate the influence of different sintering conditions and compositions on the physical properties of Si3N4/SiC composite materials.
The authors point out that samples with low defect concentration at the surface were only obtained at high-pressure conditions. The addition of SiC led only for small amounts of SIC to a dense specimen with the expected higher thermal conductivity, while higher contents of SiC hinder effective densification.
The provided values are a useful, the interpretation could be improved. I recommend a publication after minor corrections. Language should be checked carefully (abstract!). I had problems to understand the aim of this work especially at the beginning of the text.
Comments
1. The title is worded awkwardly.
Suggestion:
e.g. “(…) nanocomposites obtained by sintering at ambient and high pressure conitions”.
Abstract:
- “hipped” is slang. Define once: Hot isostatic pressed (HIPped) materials
- What is meant with matrix? The pure silicon nitride specimen or the SiN grain in the composite?
64f: Which crucibles or containers or sample holders did you use? This question is not trivial at 1800°C…
Material of the ceramic moulds?
69 Archimedes method: Which liquid? Typical weight of the specimen?
91 is “a lamp with a copper anode” correct?
143 specific heat estimated to…?
Figure 1: SE or BSE contrast?
Figure 3f: As for me, the sample consists of beta form and not, as you write, of alpha.
For XRD, the term reflections is better than bands. Bands is more suited for spectroscopy.
Figure 6: Can d/h/l/p be removed.
“According to the EDS analysis (not shown), the dark grey areas in the images of Si3N4 ceramics (Figure 6 a-h) are the silicon nitride grains.”
Please, mark them with an arrow. The “lighter” areas as well.
326: consider the purity of N2. At 1800°C traces of water/oxygen may give rise to complex gas-solid equilibria affecting the surface. Under HIP conditions, the samples are protected by the dye.
Language suggestions (there is more, please check):
40 remove: “it leads”
…can only be applied for…
45 leads
54 remove “for the research”
132 the lower surface
134 plane parallel surface,
as function of time
149 investigated/studied instead of observed
201 remove “bonding”
223 pressureless or better ambient pressure
224 was about
226 remove which is
234 sintered at
340 Remove “catastrophic”
Reviewer 2 Report
see attachment

Reviewer 3 Report
Reviewer Comment
Manuscript number: Materials- 2131313
Dear Editor,
The authors have tried to find the effect of addition of SiC into Si3N4 ceramic composite in the mechanical, thermal and physical properties of Si3N4 ceramic.Despite their effort to evaluate the effect of addition, methodology to conduct research, discuss the results, and make remarks in their research; obvious shortcomings are needed to be addressed properly.
1- First of all, the edition of the manuscript is vital, not only in terms of linguistic correction, but difficult in physical understanding. for example, lack of punctuation usage made it difficult to understand several paragraphs in this work, especially in the Abstract and introduction sections. besides, extensive English correction is essential.
2-The title should be concise but simultaneously should be in line with the Abstract section.
The proposed title does not reflect the work done at all!
This YouTube link shed a light on how to write an appropriate and attractive title:
https://www.youtube.com/watch?v=_3EfRtwM_dc&t=412s
3-several mistakes causes confusion. For example, line 21: hipping materials should be HIPing materials ! or in line 12-13: The ceramic and composite materials... whilst both ceramic and composite materials refer to both Si3N4 and Si3N4+SiC composites!
4-Abbreviation for several terms should specified from the beginning, for example, Abstract section. Just as an example, Hot Isostatic pressure (HIP),,,,,
5-The introduction section is not sufficient in terms of flow and explanation why the authors added SiC into that ceramic matrix? why they choose HIP besides conventional sintering?
6-The materials and methods section is generally acceptable. Though, several aspects should be clarified, just for example, equation 1 is the Niihara equation for the fracture toughness when Indentation Fracture (IF) technique is used. By the way, IF technique accuracy is questionable nowadays. Even we forgive authors for using this technique for the difficulty using more advanced techniques such as SEVNB,...etc, the authors need to use the term of Indentation Fracture resistance (KIFR) Instead of fracture toughness.
7-For ceramic materials, usually the term of flexural strength is used instead of bending strength? What are the difference between them? Why the authors did not use the flexural strength in their study?
8- Why the authors did not use 3-point or 4-point flexual strength techniques to measure the bending strength?
9-What is the mode of the fracture? Is it transgranular, intergranular? what are the fracture mechanisms, crack bridging, crack deflection, crack branching, pinning effect, ...etc?
10-EDX (EDS) technique is necessary to be added by the authors.
11-the FESEM micrographs in figure 6 should be clearer to justify the existence of defects and their effect on the bending strength (flexural strength).
The submitted manuscript will be accepted upon consideration these requirements.